# 40 years of research on eating disorders in domain-specific journals: Bibliometrics, network analysis, and topic modeling

**Carlos A. Almenara**[ID]*

School of Health Sciences, Universidad Peruana de Ciencias Aplicadas, Lima, Perú

* carlos.almenara@upc.pe

**Data Availability Statement:** The data that support the findings of this study are publicly available from the OSF repository: https://osf.io/5yzvd/ (DOI: 10.17605/OSF.IO/5YZVD).

## Abstract

Previous studies have used a query-based approach to search and gather scientific literature. Instead, the current study focused on domain-specific journals in the field of eating disorders. A total of 8651 documents (since 1981 to 2020), from which 7899 had an abstract, were retrieved from: International Journal of Eating Disorders (n = 4185, 48.38%), Eating and Weight Disorders (n = 1540, 17.80%), European Eating Disorders Review (n = 1461, 16.88%), Eating Disorders (n = 1072, 12.39%), and Journal of Eating Disorders (n = 393, 4.54%). To analyze these data, diverse methodologies were employed: bibliometrics (to identify top cited documents), network analysis (to identify the most representative scholars and collaboration networks), and topic modeling (to retrieve major topics using text mining, natural language processing, and machine learning algorithms). The results showed that the most cited documents were related to instruments used for the screening and evaluation of eating disorders, followed by review articles related to the epidemiology, course and outcome of eating disorders. Network analysis identified well-known scholars in the field, as well as their collaboration networks. Finally, topic modeling identified 10 major topics whereas a time series analysis of these topics identified relevant historical shifts. This study discusses the results in terms of future opportunities in the field of eating disorders.

## Introduction

There are a large and growing number of scientific publications on eating disorders (ED) [1–3]. ED are mental disorders characterized by a continuous disturbance in eating behavior, such as Anorexia Nervosa [4]. ED are usually defined according to manuals like the Diagnostic and Statistical Manual of Mental Disorders (DSM) [4]. The spectrum of ED can share some symptoms (e.g., *fear of fatness*), and these symptoms negatively impact psychosocial functioning and physical health. Due to the complexity of ED like Anorexia Nervosa, scholar literature about them covers different disciplines, such as ED related to: visual arts (e.g., art history) [5], sociology (e.g., social history) [6] and even dentistry (e.g., oral health) [7]. Thus, ED literature has a broad diversity.

**Funding:** Funding for this study was obtained from Universidad Peruana de Ciencias Aplicadas (A-006-2021).

**Competing interests:** The author has no competing interest to declare.

Previous bibliometric studies about ED have focused on: identifying the distribution by language, region and country, as well as topics and their trends [1], productivity trends and collaboration patterns [2], most cited works in Anorexia Nervosa research [8], cross-cultural aspects of ED [3], comparison of citations between types of journals [9], female authorship [10], secular trends in the scientific terminology [11, 12], the gap between scientific research and clinical practice [13], the use of keywords [14], and network analyses of common terms used in the field [15]. In particular, the current study complements the work by He et al. [1].

A standard practice of these studies is to retrieve the literature by performing a systematic search in databases like Web of Science or Scopus (i.e., employing a query-based approach), although there are some caveats worth mentioning. As noted elsewhere [16, 17], those two databases differ in journal coverage and their use can introduce bias favoring science publications (e.g., biomedicine) in detriment of arts and humanities, other than overrepresenting English-language journals. Second, databases in general (including others like PubMed, Dimensions, JSTOR), differ in their search engine functionality and information retrieval capabilities.

For example, some databases offer a controlled vocabulary like a thesaurus or taxonomy from which to choose the search terms (e.g., the Medical Subject Headings [MeSH] in PubMed), whereas others offer a full text search. Regarding the latter, indexing scanned documents to offer a full text search, requires pre-processing methods like optical character recognition (OCR), known to include typos, and post-OCR processing, both affecting information retrieval accuracy [18–23].

In other words, a query-based approach, although widely used, can be affected by several factors, including: domain expertise to design the most appropriate search strategy, the characteristics of the selected database(s), including indexation accuracy (e.g., due to OCR typos). The former is particularly important because scholars are not always consistent in using the terminology [24]. In fact, their selection of keywords is not systematic, but rather influenced by factors like their background knowledge and previous experience [25]. In this regard, within the field of ED, scholars are encouraged to use appropriate terminology [26, 27], usually a controlled vocabulary such as the Thesaurus of Psychological Index Terms. This helps to optimize the Knowledge Organization Systems (KOS) of journals and databases, such as a controlled vocabulary for information retrieval [14, 28].

In sum, most previous studies have employed a query-based search, being compelled to choose among different databases, search terms, and search strategies [29]. Nevertheless, this approach not necessarily recognizes the boundaries and limitations of both databases and we as humans interacting with machines, using diverse information retrieval strategies, and dealing with information overload [30, 31].

An alternative to the query-based approach is the one proposed in this study: to select a set of specialty journals exclusively devoted to the study of ED. Although this sampling could seem arbitrary, it was adopted: (1) to complement the findings of previous studies [1, 2] and (2) because it has in fact a sound base: the intellectual and social structure of knowledge [32–36]. We must recognize that documents need to be understood with regard to "the broader contexts in which they are produced, used, and cited" [37, p. 42]. Thus, the following sections will explain how domain-specific journals are tightly tied to an organized social and disciplinary structure. Moreover, I will explain how this approach does not necessarily exclude all ED literature from non-domain-specific journals, but rather incorporates part of it into their citations. Finally, from a complex systems perspective, I will show how domain-specific journals can be conceived as a specialized subset from the larger and more complex network comprising all ED literature.

## Domain-specific journals and its social structure

From a scientometric perspective, science, metaphorically conceived as a *knowledge space* or *knowledge landscapes*, can be defined in terms of a *network of scholars* that produce a *network of knowledge* [35]. In the former case, the social function of science has long been recognized (e.g., by Thomas Kuhn): scholars produce and communicate scientific knowledge and this organized activity has the characteristics of a social process [36, 38]. More importantly, the patterns of interactions and communication within this social organization are tightly tied, rather than isolated, to the knowledge they produce [36].

An exemplary case is the role of journal editors as gatekeepers, with studies identifying editorial gatekeeping patterns [39, 40]. According with the Network Gatekeeping Theory, inspired by the work of Kurt Lewin, gatekeeping refers to the control in the flow of information [41, 42]. In the field of ED, this intellectual and social organization of knowledge can be seen in professional societies like the Academy of Eating Disorder, which since 1981 publishes the most renowned scientific journal: The International Journal of Eating Disorders. Within its editorial board, there are distinguished scholars that can act as gatekeepers to ensure quality control and that manuscripts published by the journal are in line with the aims and scope of it.

In sum, domain-specific journals have the goal of publishing information within the boundaries of their aims and scope, allowing the diffusion of specialized knowledge.

## Domain-specific journals and its disciplinary organization

From a network perspective, specialty journals are also indicators of disciplinary organization [43], which exerts a non-trivial influence at both the global and local level of the network. To be more precise, if we visualize a network [e.g., 2, 44, 45], the local density of specialty journals evidence emerging patterns such as citation patterns by articles from the same journal or group of journals [43]. At the author level, these patterns reflect the local influence of specialty journals on scholars who adhere to their research tradition and their contributions help to advance a research agenda [46].

For example, domain-specific journals on ED often publish curated information from conferences [e.g., 47] or *special issues* about a specialized topic [e.g., 48], which commonly include a *research agenda* [48], setting the stage for future research. As we mentioned above, similar literature, such as special issues about ED published in other journals [e.g., 49], is not necessarily excluded in the analysis of domain-specific journals. Rather, such literature is commonly cited in documents from domain-specific journals and can be included in a citation analysis. Importantly, these citation patterns suggest that the former intellectual and social structure of knowledge constrains what is being studied in the future [46]. Thus, in the upcoming years, most of this specialized literature is expected to become an active *research front* [32], as evidenced by its high number of citations.

Finally, it is worth mentioning that the analysis of these patterns can reveal latent hierarchies and topological properties of journal networks. In fact, domain-specific journals can be identified through the study of the hierarchical organization of journal networks. When hierarchical network analysis is used to identify the capability of journals to spread scientific ideas, multidisciplinary journals are found at the top of the hierarchy, whereas more specialized journals are found at the bottom [50, 51]. Similarly, significant articles from a specific domain have unique topological properties that can affect the dynamic evolution of the network [52]. In sum, it is important to recognize the topological properties of networks and their latent hierarchies, both at the journal level and document level. In our case, focusing on domain-specific journals, it would be like zooming into the most central part (core) of the network topology to analyze its organization and distinctive features. Indeed, this approach is commonly

employed, for example, when studying network subsets such as niches or communities in complex systems.

## Domain-specific journals and complex adaptive systems

Domain-specific journals can also be comprehended from a complex systems standpoint, as the aggregation of the intellectual, social, and citation patterns outlined above. According to the Structural Variation Theory [53], the body of scientific knowledge can be conceived as a *complex adaptive system* (CAS). As such, it can be described and studied as a complex network with a series of characteristics like non-linearity, emergence, and self-organization; and a series of social, conceptual, and material elements that evolve over time [46]. Ideally, we must study CAS holistically to understand the properties of the system at the macrolevel [54]. In our case, this would require including all scholar literature on ED, which could be attempted using a query-based approach and employing *ad hoc* methodologies (e.g., iterative citation expansion) [45]. However, complex systems emerge from rules and behavior of lower-level components, and there is growing interest in understanding complexity from its simplest and fundamental elements and patterns [55, 56]. In our case, this can be accomplished by zooming into domain-specific patterns that emerge from the relational structure and organization of journals and papers [46], rather than focusing on the whole system which comprises all the scientific literature on ED.

This approach can be described in terms of *modularity*, a structural property of systems: the local density of specialty journals is indicative of a structural module or subsystem [57]. This property of complex systems is important because it recognizes, as we did above, the existence of subsets within networks. Indeed, scientometric studies usually attempt to detect communities based on the principle of modularity by grouping similar literature (i.e., clustering) [44, 58]. However, in the approach used in this study, rather than using bibliographic connections (e.g., through co-citation analysis) to detect domain-specific literature, we can use logical connections [59], to identify modules that operate as domain-specific representations [60]. In other words, domain-specific journals can be seen as clusters of articles that are logically linked because they all pertain to a given domain, which is explicitly stated in the aims and scope of the journals.

This modular organization has some advantages over others such as a hierarchy (e.g., Scimago categorization of journals) or a cluster obtained by literature partitioning algorithms. First, it has the advantage of reducing both *complexity bias* and *hierarchical bias*. The former is the tendency to assume and adopt a more complex system (the opposite to Occam's Razor: prefer the simplest explanation), which means to analyze all ED literature. The latter assumes that behavior is directed in a hierarchical fashion, where a central authority passes instructions to all agents in the system [54]. Second, although it still recognizes a hierarchical structure composed by diverse classes of subsystems, it assumes *heterarchy* [43, 61], which means that both hierarchical and nonhierarchical elements can be present in a system; *holarchy*, which means that systems are composed of components that can be recognized as subsystems [62]; and *glocal control*, which means that local and global phenomena in a system are achieved by local actions [63]. In simple words, sampling a set of domain-specific journals reduces complexity without affecting assumptions such as a categorical hierarchy of journals.

## The current study

To expand on previous studies [1, 2], the current study aims to answer the following research questions:

**RQ1.** Which are the most cited documents in this domain-specific corpus of articles?

**RQ2.** Which are the most important authors and their collaboration networks?

**RQ3.** Which are the most relevant topics in this domain-specific corpus of articles?

**RQ4.** How have the identified topics evolved over time (since 1981 to 2020)?

To answer these questions, this study employs a hybrid methodology. First, basic bibliometrics will be performed to identify the most cited documents. Second, network analysis will be employed to identify the most important authors and their networks of collaboration. Third, text mining, natural language processing, and machine learning algorithms will be used to identify the most relevant topics (i.e., topic modeling). Finally, a simple time series analysis will be performed to examine the evolution of these topics over time. The procedure employed for the analyses is detailed in the methods section below (and **S5 File**), whereas the dataset and the code to perform the analyses are shared in a public repository (https://doi.org/10.17605/OSF.IO/5YZVD), allowing the reproducibility of results [64].

## Methods

### Data collection

The methodology workflow is presented in **Fig 1**.

First, in May 2020, a search of journals was performed in Scimago Journal Reports (SJR, https://www.scimagojr.com/), using the term "eating disorders". In this step, the following five journals were identified: *International Journal of Eating Disorders* (ISSNs: 0276–3478, 1098-108X), *European Eating Disorders Review* (ISSNs: 1072–4133, 1099–0968), *Eating Disorders* (ISSNs: 1064–0266, 1532-530X), *Eating and Weight Disorders* (ISSNs: 1124–4909, 1590–1262), and *Journal of Eating Disorders* (ISSN: 2050-2974). The official website of each journal was then visited to confirm that the scope of the journal specifically includes the publication of research articles on eating disorders. It should be noted that the journal *Advances in Eating Disorders* (ISSNs: 2166–2630, 2166–2649) was not included because it was not found in SJR, it was published only between 2013 and 2016, it was incorporated into the journal *Eating Disorders*, and by the time of writing this article, it was not indexed neither in Scopus (https://www.scopus.com) nor in Web of Science (https://www.webofknowledge.com).

Next, also in May 2020, the Scopus database was chosen to retrieve the document records from the aforementioned journals. The election was made for no other reason than the capability of Scopus to retrieve several structured information (metadata, such as the abstract), and the file types for download are easy to manage, such as comma-separated values (CSV). Therefore, all document records published by these journals were searched in Scopus using the ISSN as the search term (e.g., *ISSN (0276–3478) OR ISSN (02763478) OR ISSN (1098-108X) OR ISSN (1098108X)*). A total of 8651 documents between 1981 and 2020 were retrieved (of which 7899

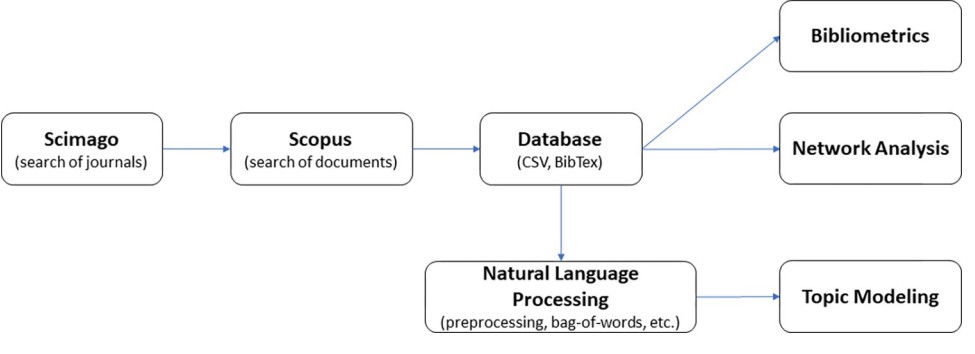

**Fig 1. Workflow of the methodology used in this research study.**

had an abstract): 4185 (48.38%) from the International Journal of Eating Disorders, 1540 (17.80%) from Eating and Weight Disorders, 1461 (16.88%) from the European Eating Disorders Review, 1072 (12.39%) from Eating Disorders, and 393 (4.54%) from the Journal of Eating Disorders. These 8651 documents included a total of 213,744 references. It should be noted that the International Journal of Eating Disorders is the oldest of these journals, established in 1981. The S7 and S8 Files provide the number of documents per year and per journal. The document records were downloaded from Scopus both as comma separated values (CSV) and as BibTex (http://www.bibtex.org/), and selecting all fields available (i.e., title, author, abstract, etc.). Due to copyright, the full text of all documents was not retrieved but rather their metadata (i.e, title, author, date, abstract), whilst the dataset shared online (https://doi.org/10.17605/OSF.IO/5YZVD) is the one obtained after the preprocessing procedures detailed below.

Analyses were performed using open software: R Statistical Software 4.0.3 (Bunny-Wunnies Freak Out) [65], and Python programming language version 3.9.1 (https://www.python.org/).

## Bibliometric analysis and network analysis in R

The *biblioshiny* application from the R package *bibliometrix* [66] was used to preprocess the CSV file. Next, it was used to identify the most cited documents. Local citations (i.e., citations only from documents whithin the dataset), and global citations (i.e., citations made by any document from the whole Scopus database), were computed. Biblioshiny was also used for network analysis as described by Batagelj & Cerinšek [67], and Aria & Cuccurullo [66]. Regarding the network, it is defined as a pair of sets: a set of nodes or vertices and a set of edges (link between nodes) [68]. In this study, when authors were treated as nodes, a link would represent co-authorship or collaboration [see 69]. More precisely, the Louvain algorithm for community detection [70] was used to identify communities within the collaboration network. This algorithm identifies densely connected nodes within the network (i.e., communities) [e.g., 71]. It works unconstrained to automatically extract a number of clusters although it requires basic network parameters as input. These network parameters were: up to 100 nodes, a minimum of two edges by node, and the removal of isolated nodes. For network layout visualization, the Fruchterman & Reingold [72] algorithm was chosen. Finally, common centrality measures were calculated: betweenness, closeness, and PageRank. *Betweenness centrality* refers to "the frequency that a node is located in the shortest path between other nodes" [73, p. 772]. *Closeness centrality* refers to nodes that can easily reach others in the network, whilst *PageRank*, originally created to rank websites [74], has been used to rank authors because it takes into account the weight of influential nodes [75].

## Topic modeling: Dimensionality reduction and matrix factorization

As can be seen in the workflow (**Fig 1**), once network analysis was finished, a series of steps (detailed in S5 File) were necessary to preprocess the dataset prior to topic modeling. Topic modeling refers to applying machine learning techniques to find topics by extracting semantic information from unstructured text in a corpus [76]. As we explain in **S5 File**, to this point we end up with a high-dimensional and sparse document-term matrix. In other words, we have many features (columns) each corresponding to a term in our corpus, and for a given document (rows) we have many columns with zero values meaning the term of that column is not in the given document. To deal with sparsity, we can perform *dimensionality reduction* to obtain a representation that effectively captures the variability in the data. In summary, dimensionality reduction can be categorized in *feature extraction* and *feature selection*; the former combines the original feature space into a new one, whereas the latter selects a subset of features [77].

As explained in S5 File, the term frequency (TF) and the term frequency-inverse document frequency (TF-IDF) were used as feature extraction for vectorization. Then, the following machine learning algorithms were applied for topic modeling: Latent Dirichlet Allocation (LDA) [78], Latent Semantic Analysis (LSA or Latent Semantic Indexing) [79], Hierarchical Dirichlet Process (HDP) [80], and Non-negative Matrix Factorization (NMF) [81]. LDA is a generative probabilistic model that decomposes the document-term matrix into a topic-term matrix and a document-topic matrix, and it is commonly used for topic discovering from a corpus [e.g., 82]. LSA utilizes a truncated Singular Value Decomposition for decomposition and can work efficiently on TF or TF-IDF sparse matrices. In a fully unsupervised framework, the HDP model is characterized by inferring the number of topics on its own. Finally, NMF is an alternative approach that implements the Nonnegative Double Singular Value Decomposition, an algorithm suitable for sparse factorization [83].

First, the GENSIM library [84] was used for topic modeling because it provides a way to calculate *topic coherence*, an index to compare models based on measures of segmentation, probability estimation, confirmation measure, and aggregation [see 85]. Therefore, based on a TF matrix, HDP, LSA, NMF, and LDA were performed in GENSIM and compared in topic coherence. Once identified the topic modeling algorithms with the highest topic coherence, *scikit-learn* [86] was used because it provides an Exhaustive Grid Search option for ensemble learning the models (i.e., automatically fine-tuning the parameters to find the most optimal). Finally, once the topics were extracted, a simple time series analysis was performed to visualize the changes over time in the topics found. This analysis consisted of simply plotting the number of documents for each topic across years, from 1981 to 2020.

## Results

First, bibliometric analyses were performed to identify the most cited documents. Local citations are presented in **Table 1** (and the S1 File), whereas global citations are in **Table 2** (and the S2 File).

Next, a network analysis was performed to identify the most important authors (**Table 3**) and their collaboration networks (**Fig 2**, see also **S3 File**, a dataset, and **S4 File**, an interactive visualization in HTML and JavaScript, also available online: https://osf.io/5yzvd/). This collaboration network analysis identified eight clusters with 96 authors: (1) red color, 4 authors; (2) blue, 15 authors; (3) green, 17 authors; (4) purple 21 authors; (5) orange, 2 authors; (6) brown, 18 authors; (7) pink, 2 authors; (8) grey, 17 authors.

Regarding the most relevant topics, LDA and NMF were superior to HDP and LSA in topic coherence. Then, when ensemble learning was used for LDA (based on TF) and NMF (based on TF-IDF), NMF provided the most meaningful results, and 10 topics were identified (**Table 4**).

The labels for the topics were manually added based on the top 10 keywords and their respective weights. Thus, each topic was manually labeled as follows: (1) risk factors for eating disorders, (2) body image dissatisfaction, (3) Binge Eating Disorder diagnosis, (4) weight loss, weight control, and diet, (5) clinical groups, (6) treatment outcome, (7) family and parent-child, (8) binge and purge episodes, (9) gender and subgroups, (10) EDNOS.

To examine how these topics have evolved over time, a simple time series analysis plot was created (**Fig 3** and S6 File).

## Discussion

This study analyzed 8651 documents between 1981 and 2020 from domain-specific journals in the field of eating disorders. The aims were: to identify the most cited documents, the most

**Table 1. Top 10 most cited documents according to local citations (within the dataset of documents)[a].**

| # | Reference | Local Citations | Global Citations |
|---|-----------|-----------------|------------------|
| 1 | Garner, D. M., Olmstead, M. P., & Polivy, J. (1983). Development and validation of a multidimensional eating disorder inventory for anorexia nervosa and bulimia. *International Journal of Eating Disorders*, *2*(2), 15–34. | 573 | 3128 |
| 2 | Cooper, P. J., Taylor, M. J., Cooper, Z., & Fairburn, C. G. (1987). The development and validation of the Body Shape Questionnaire. *International Journal of Eating Disorders*, *6*(4), 485–494. | 216 | 1251 |
| 3 | Hoek, H. W., & van Hoeken, D. (2003). Review of the prevalence and incidence of eating disorders. *International Journal of Eating Disorders*, *34*(4), 383–396. | 157 | 972 |
| 4 | Strober, M., Freeman, R., & Morrell, W. (1997). The long-term course of severe anorexia nervosa in adolescents: Survival analysis of recovery, relapse, and outcome predictors over 10–15 years in a prospective study. *International Journal of Eating Disorders*, *22*(4), 339–360. | 139 | 674 |
| 5 | Spitzer, R. L., Yanovski, S., Wadden, T., Wing, R., Marcus, M. D., Stunkard, A., . . . Horne, R. L. (1993). Binge eating disorder: Its further validation in a multisite study. *International Journal of Eating Disorders*, *13*(2), 137–153. | 131 | 668 |
| 6 | Berg, K. C., Peterson, C. B., Frazier, P., & Crow, S. J. (2012). Psychometric evaluation of the eating disorder examination and eating disorder examination-questionnaire: A systematic review of the literature. *International Journal of Eating Disorders*, *45*(3), 428–438. | 126 | 410 |
| 7 | Spitzer, R. L., Devlin, M., Walsh, B. T., Hasin, D., Wing, R., Marcus, M., . . . Nonas, C. (1992). Binge eating disorder: A multisite field trial of the diagnostic criteria. *International Journal of Eating Disorders*, *11*(3), 191–203. | 123 | 647 |
| 8 | van Strien, T., Frijters, J. E. R., Bergers, G. P. A., & Defares, P. B. (1986). The Dutch Eating Behavior Questionnaire (DEBQ) for assessment of restrained, emotional, and external eating behavior. *International Journal of Eating Disorders*, *5*(2), 295–315. | 107 | 1846 |
| 9 | Berkman, N. D., Lohr, K. N., & Bulik, C. M. (2007). Outcomes of eating disorders: A systematic review of the literature. *International Journal of Eating Disorders*, *40*(4), 293–309. | 79 | 341 |
| 10 | Bryant-Waugh, R. J., Cooper, P. J., Taylor, C. L., & Lask, B. D. (1996). The use of the eating disorder examination with children: A pilot study. *International Journal of Eating Disorders*, *19*(4), 391–397. | 78 | 319 |

*Note*.

[a] Local citations are citations only from documents whithin the dataset.

important authors and their collaboration networks, and the most relevant topics and their evolution over time. The results expand previous findings of studies that employed a query-based approach and included articles dating back as far as 1900 [13]. In particular the results expand the studies by Jinbo He et al. (2022) and Juan-Carlos Valderrama-Zurián, et al. (2017), which employed a similar methodology [1, 2]. For example, He et al. (2022) created a collaboration network, although it was based on countries rather than authors [1]. Therefore, the results obtained here (e.g., author centrality measures, author clusters) provide a more fine grained understanding of the relevance and contribution of individual authors and their collaboration networks. Furthermore, He et al. (2022) [1] identified top authors based on traditional performance metrics (e.g., h-index), and it should be noted that there is some criticism towards their use and a claim to shift towards more responsible metrics of research excellence [87]. Then, He et al. (2022) [1] employed LDA for topic modeling, whilst this study employed NMF. Although LDA is largely used, in this study NMF outperformed LDA in interpretability, reproducibility, and as we said above, it suits better for short texts, as is the case of article abstracts used here. Finally, the top journals identified by He et al. (2022) confirmed that the

**Table 2. Top 10 most cited references within the dataset (global citations)[a].**

| # | Reference | Global Citations |
|---|-----------|------------------|
| 1 | Garner, D. M., Olmstead, M. P., & Polivy, J. (1983). Development and validation of a multidimensional eating disorder inventory for anorexia nervosa and bulimia. *International Journal of Eating Disorders*, *2*(2), 15–34. | 3128 |
| 2 | Fairburn, C. G., & Beglin, S. J. (1994). Assessment of eating disorders: Interview or self-report questionnaire? *International Journal of Eating Disorders*, *16*(4), 363–370. | 2735 |
| 3 | van Strien, T., Frijters, J. E. R., Bergers, G. P. A., & Defares, P. B. (1986). The Dutch Eating Behavior Questionnaire (DEBQ) for assessment of restrained, emotional, and external eating behavior. *International Journal of Eating Disorders*, *5*(2), 295–315. | 1846 |
| 4 | Cooper, P. J., Taylor, M. J., Cooper, Z., & Fairburn, C. G. (1987). The development and validation of the Body Shape Questionnaire. *International Journal of Eating Disorders*, *6*(4), 485–494. | 1251 |
| 5 | Hoek, H. W., & van Hoeken, D. (2003). Review of the prevalence and incidence of eating disorders. *International Journal of Eating Disorders*, *34*(4), 383–396. | 972 |
| 6 | Groesz, L. M., Levine, M. P., & Murnen, S. K. (2002). The effect of experimental presentation of thin media images on body satisfaction: A meta-analytic review. *International Journal of Eating Disorders*, *31*(1), 1–16. | 960 |
| 7 | Cooper, Z., & Fairburn, C. G. (1987). The Eating Disorder Examination: A semi-structured interview for the assessment of the specific psychopathology of eating disorders. *International Journal of Eating Disorders*, *6*(1), 1–8. | 848 |
| 8 | Strober, M., Freeman, R., & Morrell, W. (1997). The long-term course of severe anorexia nervosa in adolescents: Survival analysis of recovery, relapse, and outcome predictors over 10–15 years in a prospective study. *International Journal of Eating Disorders*, *22*(4), 339–360. | 674 |
| 9 | Spitzer, R. L., Yanovski, S., Wadden, T., Wing, R., Marcus, M. D., Stunkard, A., . . . Horne, R. L. (1993). Binge eating disorder: Its further validation in a multisite study. *International Journal of Eating Disorders*, *13*(2), 137–153. | 668 |
| 10 | Spitzer, R. L., Devlin, M., Walsh, B. T., Hasin, D., Wing, R., Marcus, M., . . . Nonas, C. (1992). Binge eating disorder: A multisite field trial of the diagnostic criteria. *International Journal of Eating Disorders*, *11*(3), 191–203. | 647 |

*Note*.

[a] Global citations are citations made by any document from the whole Scopus database.

**Table 3. Indices of centrality for the top 10 authors in the network analysis of collaboration.**

| # | Author | Betweenness | Closeness[a] | PageRank[b] | Cluster |
|---|--------|-------------|--------------|-------------|---------|
| 1 | Ross D Crosby | 714.15 | 0.18 | 0.35 | 8 |
| 2 | James E Mitchell | 712.74 | 0.18 | 0.39 | 8 |
| 3 | Janet Treasure | 711.41 | 0.18 | 0.29 | 6 |
| 4 | Cynthia M Bulik | 553.07 | 0.18 | 0.31 | 4 |
| 5 | Daniel Le Grange | 303.62 | 0.18 | 0.25 | 8 |
| 6 | Walter H Kaye | 192.64 | 0.18 | 0.21 | 4 |
| 7 | Katherine A Halmi | 80.31 | 0.18 | 0.19 | 4 |
| 8 | Stephen A Wonderlich | 60.54 | 0.17 | 0.25 | 8 |
| 9 | Scott J Crow | 53.8 | 0.17 | 0.23 | 8 |
| 10 | Carol B Peterson | 2.44 | 0.17 | 0.22 | 8 |

*Note*. The order of authors is sorted by the value in betweenness centrality.

[a] Original closeness values were multiplied by 100 to display values with only two decimals.

[b] Original PageRank values were multiplied by 10 to display values with only two decimals

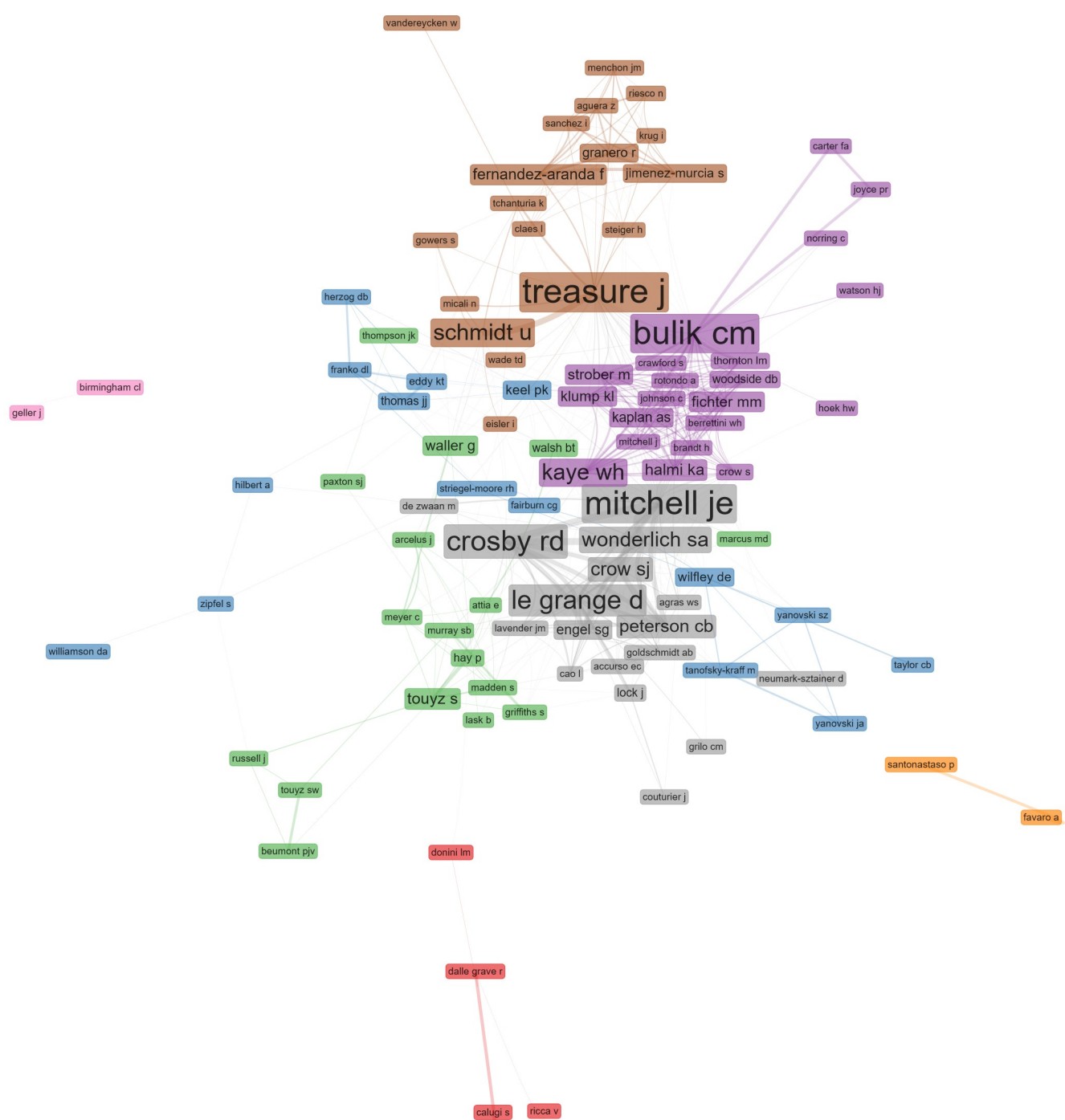

**Fig 2. Network of collaboration including 96 authors and eight clusters.**

five journals selected for this study are in fact among the most important in the field of eating disorders [1]. In the case of Valderrama-Zurián, et al. (2017) [2], they also focused on authors' productivity trends whereas their social network analysis was focused on network metrics such as the number of nodes and edges over time, which precludes to inspect the social

**Table 4. Results of topic modeling: First nine keywords of each topic with their respective weight below each keyword.**

| Topic | KW-1 | KW-2 | KW-3 | KW-4 | KW-5 | KW-6 | KW-7 | KW-8 | KW-9 | Documents |
|---|---|---|---|---|---|---|---|---|---|---|
| 1 | eat | disorder | eating | symptom | self | score | high | factor | risk | 2809 |
| | 2.41 | 2.29 | 1.16 | 0.91 | 0.83 | 0.8 | 0.7 | 0.69 | 0.65 | |
| 2 | body | image | dissatisfaction | size | disturbance | ideal | esteem | girl | appearance | 1136 |
| | 3.73 | 1.98 | 1.46 | 0.67 | 0.44 | 0.44 | 0.43 | 0.42 | 0.4 | |
| 3 | bed | obese | binge | individual | disorder | obesity | criterion | subject | psychopathology | 928 |
| | 4.09 | 0.56 | 0.48 | 0.28 | 0.27 | 0.23 | 0.22 | 0.19 | 0.18 | |
| 4 | weight | loss | bmi | overweight | gain | obese | obesity | normal | control | 735 |
| | 3.95 | 1.16 | 0.9 | 0.73 | 0.71 | 0.68 | 0.67 | 0.49 | 0.48 | |
| 5 | patient | nervosa | anorexia | bulimia | case | disorder | group | clinical | bulimic | 671 |
| | 3.48 | 2.04 | 1.79 | 1.0 | 0.81 | 0.54 | 0.51 | 0.43 | 0.43 | |
| 6 | treatment | outcome | therapy | cbt | intervention | change | program | follow | improvement | 379 |
| | 3.34 | 1.12 | 0.8 | 0.79 | 0.64 | 0.62 | 0.62 | 0.58 | 0.47 | |
| 7 | child | family | parent | mother | adolescent | parental | maternal | girl | fbt | 356 |
| | 2.89 | 2.0 | 1.51 | 1.12 | 0.89 | 0.57 | 0.48 | 0.46 | 0.32 | |
| 8 | binge | eating | episode | eat | purge | food | frequency | eater | Behavior | 328 |
| | 3.8 | 1.17 | 1.01 | 0.5 | 0.49 | 0.46 | 0.46 | 0.45 | 0.39 | |
| 9 | woman | man | bulimic | group | bulimia | report | black | white | pregnancy | 282 |
| | 4.12 | 1.69 | 0.76 | 0.61 | 0.51 | 0.39 | 0.37 | 0.36 | 0.34 | |
| 10 | bn | ednos | bulimia | nervosa | pd | lifetime | disorder | purge | rate | 275 |
| | 4.02 | 0.53 | 0.51 | 0.39 | 0.19 | 0.18 | 0.17 | 0.17 | 0.15 | |

*Note.* KW = Keyword. Numbers below each keyword indicate their weight within the topic they belong.

network at the author level. Therefore, this study also expands on the findings of Valderrama-Zurián, et al. (2017) [2].

Below, we discuss in more detail the results of the analysis employed to answer the four research questions outlined in the introduction.

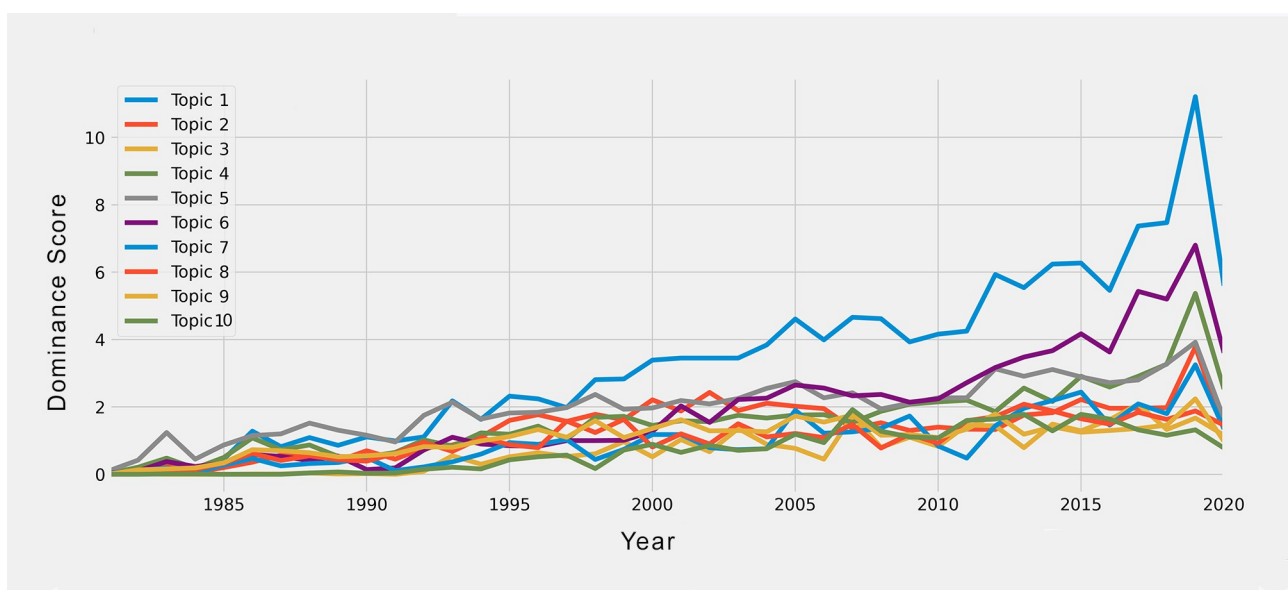

**Fig 3. Time series showing the topic dominance scores for the ten topics found.** *Note.* Values in the y-axis are the sum of the weight values from the NMF analysis for topic dominance, per year and per topic. Values go from minimum 0 to maximum 11.2 (see S6 File).

## Bibliometric analysis

The top cited documents were all from the International Journal of Eating Disorders. As noted above, this journal is the oldest one (it started in 1981), and it has the largest number of articles per year, with the exception of the year 2019 when it was outperformed by the Eating and Weight Disorders journal (see S7 and S8 Files). The majority of top cited documents were related to the development of instruments for the assessment of eating disorders or the course and outcome of eating disorders. For example, we can see in the results the most common instruments used for the screening of eating disorders, as well as the evaluation of its core symptoms: Eating Disorder Inventory (EDI), Body Shape Questionnaire (BSQ), Dutch Eating Behavior Questionnaire (DEBQ), and Eating Disorder Examination Questionnaire (EDE-Q). These instruments are widely used to screen the general population, as well as in clinical settings, together with more recent instruments [88]. It should be noted, however, that in clinical practice settings the use of instruments for the diagnosis and the different phases of the treatment process is not necessarily widespread [89, 90]. To reduce this gap, some authors suggest to provide assessment training and/or assessment guidelines for mental health professionals and general practitioners in primary health care [91, 92]. This can help obtain a comprehensive clinical assessment, particularly of individuals with higher risk such as young adolescents with restrictive Anorexia Nervosa [93]. The instruments mentioned above are reliable measures, and they could be used online for a quick screening or session by session for ongoing monitoring, although further research is necessary [e.g., 94–96].

The rest of most cited documents include important review articles on epidemiology (Hoek & van Hoeken, 2003, in **Table 1**); the course and outcome of eating disorders (Berkman, Lohr & Bulik, 2007; Strober, Freeman & Morrell, 1997; in **Table 1**); and the diagnosis of Binge Eating Disorder (Spitzer et al., 1992, 1993, in **Table 1**). These results are similar to previous studies in which measurement methods (including instrument development), epidemiology, and review articles were the most common type of document [8, 9].

Finally, the large number of articles on the diagnosis of Binge Eating Disorder, which was not fully recognized as a mental disorder in the Diagnostic and Statistical Manual of Mental Disorders (DSM) until its fifth edition [4], reveal that the recognition of Binge Eating Disorder as an own disorder took several years. To reach expert consensus in a shorter time, eating disorder professionals should pay special attention to emerging eating problems, such as Orthorexia Nervosa [97].

## Network analysis

The network analysis identified eight clusters with 96 authors. Previous studies have examined the network of authors in the field in terms of network statistics such as number of edges or network density [2]. By contrast, this study provides a more fine-grained network analysis, identifying experts and group of experts in the field of eating disorders. As seen in the results section, the majority are distinguished authors with contributions dating back to the early 1980s.

The author with the largest betweenness centrality was Ross D Crosby (Sanford Center for Biobehavioral Research, United States), followed by James E Mitchell (University of North Dakota, United States) which has the largest value in PageRank. Authors with high betweenness centrality can act as both enablers and gatekeepers of information flow between communities [75]. Moreover, it has been found that authors with high betweenness centrality establish more collaborations than those high in closeness centrality [75]. In summary, the results of centrality measures can help to identify experts in the field of eating disorders, particularly

authors that can quickly reach other authors in the network (high in closeness), act as gate-keepers (high in betweenness), or relate to influential others (high in PageRank).

Regarding the clusters identified by the network analysis, in the same cluster of Ross D Crosby and James E Mitchell are found other renowned authors like Daniel Le Grange (University of California, San Francisco, United States), Stephen A Wonderlich (Sanford Center for Biobehavioral Research, United States), and Carol B Peterson (University of Minnesota, United States). Among the most relevant results of collaboration of this cluster we can find studies on the ecological momentary assessment of eating disorders [98], the psychometric properties of the EDE-Q [99], and the diagnosis of Binge Eating Disorder [100].

The second largest cluster includes authors like Cynthia M Bulik (University of North Carolina at Chapel Hill, United States), Walter H Kaye (University of California, San Diego, United States), and Katherine A Halmi (Weill Cornell Medical College, United States). The results of their collaboration include studies related to the phenotypic characterization of eating disorders, such as the International Price Foundation Genetic Study, a multisite study that included a large sample of patients with eating disorders and their families [e.g., 101].

Finally, the third largest cluster includes authors like Janet Treasure (King's College London, England), Ulrike Schmidt (King's College London, England), and Tracey D Wade (Flinders University, Australia), which are widely recognized by the Maudsley Model for Treatment of Adults with Anorexia Nervosa (MANTRA) [102, 103]. Interestingly, this is the only cluster that includes collaborations with authors from non-English speaking countries, more specifically from Spain. Examples of these collaborations include studies resulting from the Wellcome Trust Case Control Consortium 3 (WTCCC3) and the Genetic Consortium for AN (GCAN) [104], and other studies with clinical samples in Spain [e.g., 105].

On the other hand, the results reveal the importance of multisite studies that strengthen collaboration and originate in relevant outcomes for the prevention and treatment of eating disorders. Research groups could look for opportunities to collaborate in multisite studies and strengthen both their interdisciplinary and transdisciplinary collaboration, and their collaboration with less common partners such as stakeholders and policy makers [106, 107]. By establishing these integrative and strategic collaborations we can promote translational research, and thus helping to reach broader public health goals [108].

## Topic modeling

The combination of TF-IDF and NMF provided meaningful results, identifying 10 topics. After labeling these topics based on the first 10 keywords and their respective weights, we can see that most of the research on eating disorders done in the past 40 years has focused on their prevention and treatment. Interestingly, the time trend analysis of these topics revealed a noticeable change in the first *lustrum* of the 1990s. Whereas during the early 1980s the study of clinical groups (topic 5) was the most dominant topic, from the mid-1990s, this topic was surpassed by the study of risk factors of eating disorders (topic 1). This indicates an increasing interest for the prevention rather than solely the treatment of eating disorders. This result is consistent with the historical shift that occurred in the United States when in 1992 the Institute of Medicine (IOM) Committee on Prevention of Mental Disorders was created [109]. Then two years later, a report on reducing risk factors for mental disorders and promoting a preventive approach in research was published [110]. As expected, this shift had echo in several scholars at the time, became a research front, and relevant publications started to include more information on the prevention of eating disorders, including a special issue [111], book chapters [112], and progressively entire books [113]. It is important to note that this historical shift, as well as later others like in 2017 [114], were favorable, because in other cases like obesity, it

took more time to focus on its prevention due to different issues, including the pressure of the weight loss industry and its commercial interest [115].

Another interesting finding was that the outcome of the treatment of eating disorders (topic 6), is the second most important topic of 2013, and this finding has important aspects to discuss. First, the surge of state-of-the-art machine learning algorithms provide several opportunities to build intelligent systems for precision medicine. Thus, the treatment course and outcome of eating disorders can be more personalized, guided, and enhanced with the help of predictive technologies and intelligent systems [e.g., 116]. Second, as suggested elsewhere [117], the advantages of technology can be particularly relevant for certain age groups like adolescents, and when a digital intervention is employed [118]. In summary, treatment outcome is currently an important topic, and future studies can deploy digital interventions and machine learning algorithms for a more precise treatment planning.

## Limitations and conclusions

Although this study has strengths, such as using data and code that allows the reproducibility of the results, readers should consider some limitations. First, the analysis of most cited documents is for all the time span, and more recent highly cited documents are underrepresented. Moreover, the journal *Advances in Eating Disorders* was not included due to indexing issues. Nevertheless, this study provides the code and a detailed procedure to allow researcher to perform further analyses, such as document co-citation analysis. Future studies can also evaluate the *Mexican Journal of Eating Disorders* (*Revista Mexicana de Trastornos Alimentarios*, ISSN 2007-1523), which has published articles primarily in Spanish [119]. Second, the network analysis included close to 100 scholars mostly with a long trajectory in the field, and this can be a limitation in representing more younger scientists or newcomers [2]. Future studies can focus on a larger number of scholars and apply different techniques in network analysis, such as other community detection techniques [e.g., 120]. Finally, the results of topic modeling suggested a solution of 10 topics out of up to 30 topics solution models tested. Although there is not a universally accepted approach to establish the number of topics, this study relied on several strategies, including ensemble learning, to automatically fine-tune the parameters of the machine learning algorithms, stability, and heuristic approaches [121]. Future studies can try other machine learning algorithms and techniques to retrieve topics [121].

In conclusion, this study analyzed 40 years of research on eating disorders, identified the most cited articles, networks of collaboration, experts in the field, and the 10 major topics in the field.

## Supporting information

**S1 File. Most local cited documents.**
(CSV)

**S2 File. Most global cited documents.**
(CSV)

**S3 File. Network statistics.**
(CSV)

**S4 File. Network of collaboration including close to one hundred authors.**
(HTML)

**S5 File. Data preprocessing and text representation in Python.**
(DOCX)

**S6 File. Sum of NMF results for topic dominance per year and per topic.**
(CSV)

**S7 File. Number of documents per year and per journal.**
(CSV)

**S8 File. Trends over time in number of documents per journal.**
(TIFF)

## Author Contributions

**Conceptualization:** Carlos A. Almenara.

**Data curation:** Carlos A. Almenara.

**Formal analysis:** Carlos A. Almenara.

**Investigation:** Carlos A. Almenara.

**Methodology:** Carlos A. Almenara.

**Project administration:** Carlos A. Almenara.

**Resources:** Carlos A. Almenara.

**Software:** Carlos A. Almenara.

**Visualization:** Carlos A. Almenara.

**Writing – original draft:** Carlos A. Almenara.

**Writing – review & editing:** Carlos A. Almenara.

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
