## [Decision Letter · Decision Letter 0]

17 Jun 2021

PONE-D-21-04053

40 years of research on eating disorders: Bibliometrics, network analysis, and topic modeling

PLOS ONE

Dear Dr. Almenara,

Thank you for submitting your manuscript to PLOS ONE. After careful consideration, we feel that it has merit but does not fully meet PLOS ONE’s publication criteria as it currently stands. Therefore, we invite you to submit a revised version of the manuscript that addresses the points raised during the review process.

Please carefully consider all the concerns raised by #Reviewer 1.

The acceptance of the paper will require a solid justification of  the adoption of a narrow definition of 'eating disorders literature', a major issue highlighted by both reviewers.

Consequently, you should also introduce more cautions in the discussion of the overall relevance of  your work.

We look forward to receiving your revised manuscript.

Kind regards,

Alberto Baccini, Ph.D.

Academic Editor

PLOS ONE

Journal Requirements:

Additional Editor Comments (if provided):

Reviewers' comments:

Reviewer's Responses to Questions

**Comments to the Author**

1. Is the manuscript technically sound, and do the data support the conclusions?

Reviewer #1: No

Reviewer #2: Yes

2. Has the statistical analysis been performed appropriately and rigorously? 

Reviewer #1: Yes

Reviewer #2: Yes

3. Have the authors made all data underlying the findings in their manuscript fully available?

Reviewer #1: Yes

Reviewer #2: Yes

4. Is the manuscript presented in an intelligible fashion and written in standard English?

Reviewer #1: No

Reviewer #2: Yes

5. Review Comments to the Author

Reviewer #1: Please, see the attached file "Review Comments to the Author.pdf"

Reviewer #2: The present manuscript presents a rather comprehensive but superficial analysis of three interrelated indexes of the eating disorders literature. I have two main criticisms. The first is that I’m not convinced or do not quite understand why the author focused on just papers published in eating-disorders specialty journals. There could be a good reason to do so—but they explanation given was unconvincing. I would argue that the most “central” or “key” works are those that, regardless of where they are published, reach the broader audiences and are the most highly cited globally. My second criticism is that there is little effort to interrelate or integrate the results from the three analyses—that is, what is the nature of the relationship between being the most cited article, the most productive-impactful network, and the topic of the work?

Overall, the data analyses and the results are interesting, but the author needs to do a better job at explaining what the work attempts to do and why, the extent to which the mission was accomplished, and what limitations the current work has—including its narrow focus on literature published in “specialty” journals.

6. PLOS authors have the option to publish the peer review history of their article (what does this mean?). If published, this will include your full peer review and any attached files.

Reviewer #1: No

Reviewer #2: No

---

## [Author Response · Author response to Decision Letter 0]

10 May 2022

Please see the attached document "Responses_to_Reviewers.pdf"

---

## [Decision Letter · Decision Letter 1]

27 Jun 2022

PONE-D-21-04053R140 years of research on eating disorders in domain-specific journals: Bibliometrics, network analysis, and topic modelingPLOS ONE

Dear Dr. Almenara,

Thank you for submitting your manuscript to PLOS ONE. After careful consideration, we feel that it has merit but does not fully meet PLOS ONE’s publication criteria as it currently stands. Therefore, we invite you to submit a revised version of the manuscript that addresses the points raised during the review process.

We look forward to receiving your revised manuscript.

Kind regards,

Alberto Baccini, Ph.D.

Academic Editor

PLOS ONE

Journal Requirements:

Reviewers' comments:

Reviewer's Responses to Questions

**Comments to the Author**

1. If the authors have adequately addressed your comments raised in a previous round of review and you feel that this manuscript is now acceptable for publication, you may indicate that here to bypass the “Comments to the Author” section, enter your conflict of interest statement in the “Confidential to Editor” section, and submit your "Accept" recommendation.

Reviewer #1: (No Response)

Reviewer #2: All comments have been addressed

2. Is the manuscript technically sound, and do the data support the conclusions?

Reviewer #1: Partly

Reviewer #2: Yes

3. Has the statistical analysis been performed appropriately and rigorously? 

Reviewer #1: Yes

Reviewer #2: Yes

4. Have the authors made all data underlying the findings in their manuscript fully available?

Reviewer #1: Yes

Reviewer #2: Yes

5. Is the manuscript presented in an intelligible fashion and written in standard English?

Reviewer #1: Yes

Reviewer #2: Yes

6. Review Comments to the Author

Reviewer #1: See attached file

Reviewer #2: I find the author was very responsive to the reviewer's comments and critiques. Whereas the sampling method could be questioned, I believe the data are valid and reliable for the scope of journals included. Of all the questions addressed, the most novel and substantive contribution is the analysis of topic trends over time-this alone makes the manuscript worthy of publication.

7. PLOS authors have the option to publish the peer review history of their article (what does this mean?). If published, this will include your full peer review and any attached files.

Reviewer #1: No

Reviewer #2: No

---

## [Author Response · Author response to Decision Letter 1]

9 Oct 2022

Please see attached PDF file "Responses to Reviewer".

---

## [Decision Letter · Decision Letter 2]

15 Nov 2022

PONE-D-21-04053R240 years of research on eating disorders in domain-specific journals: Bibliometrics, network analysis, and topic modelingPLOS ONE

Dear Dr. Almenara,

Thank you for submitting your manuscript to PLOS ONE. After careful consideration, we feel that it has merit but does not fully meet PLOS ONE’s publication criteria as it currently stands. Therefore, we invite you to submit a revised version of the manuscript that addresses the points raised during the review process. I have a couple of  very minor points that I Think you should fix before the paper is accepted for publication.

1. The description of a network is not standard. You wrote: . "In network analysis, there are two main components: nodes (vertices) and edges (links between nodes)". Here you are mixing the notion of network with the analysis of networks. Moreover "components"  is a technical term in network analysis: better to avoid confusion. So, sraphs or networks are the object studied by network analysis. A network is defined as a a pair of sets: a set of nodes  or vertices and a set of edges (link between nodes). 

2. In describing the methodology you wrote:  "More precisely, the Louvain algorithm for community detection [70] was used to create a collaboration network. This algorithm identifies densely connected nodes within the network (i.e., communities) [e.g., 71]". Please note that the collaboration network is not created by the Louvain algorithm. The Louvain algorithm is used for identyfying communities in the collaboration network.  

We look forward to receiving your revised manuscript.

Kind regards,

Alberto Baccini, Ph.D.

Academic Editor

PLOS ONE

Journal Requirements:

Reviewers' comments:

Reviewer's Responses to Questions

**Comments to the Author**

1. If the authors have adequately addressed your comments raised in a previous round of review and you feel that this manuscript is now acceptable for publication, you may indicate that here to bypass the “Comments to the Author” section, enter your conflict of interest statement in the “Confidential to Editor” section, and submit your "Accept" recommendation.

Reviewer #2: All comments have been addressed

2. Is the manuscript technically sound, and do the data support the conclusions?

Reviewer #2: Yes

3. Has the statistical analysis been performed appropriately and rigorously? 

Reviewer #2: Yes

4. Have the authors made all data underlying the findings in their manuscript fully available?

Reviewer #2: Yes

5. Is the manuscript presented in an intelligible fashion and written in standard English?

Reviewer #2: Yes

6. Review Comments to the Author

Reviewer #2: Reviewer 1 was amazingly generous in providing very detailed and good guiding recommendations. I also believe the authors have been highly accommodating and the manuscript has improved substantively. Thus, I continue to be supportive of accepting the manuscript for publication.

7. PLOS authors have the option to publish the peer review history of their article (what does this mean?). If published, this will include your full peer review and any attached files.

Reviewer #2: **Yes: **Antonio Cepeda-Benito

---

## [Author Response · Author response to Decision Letter 2]

24 Nov 2022

Reviewers didn't make any further comments.

---

## [Editor Report · Decision Letter 3]

29 Nov 2022

40 years of research on eating disorders in domain-specific journals: Bibliometrics, network analysis, and topic modeling

PONE-D-21-04053R3

Dear Dr. Almenara,

We’re pleased to inform you that your manuscript has been judged scientifically suitable for publication and will be formally accepted for publication once it meets all outstanding technical requirements.

Kind regards,

Alberto Baccini, Ph.D.

Academic Editor

PLOS ONE
---

## [Editor Report · Acceptance letter]

6 Dec 2022

PONE-D-21-04053R3 

40 years of research on eating disorders in domain-specific journals: Bibliometrics, network analysis, and topic modeling 

Dear Dr. Almenara:

I'm pleased to inform you that your manuscript has been deemed suitable for publication in PLOS ONE. Congratulations! Your manuscript is now with our production department. 

Kind regards, 

on behalf of

Prof. Alberto Baccini 

Academic Editor

PLOS ONE